Recognition of emotion and pain by owners benefits the welfare of donkeys in a challenging working environment

Bukhari Syed S.U.H. habukhari2-c@my.cityu.edu.hk 1 2
McElligott Alan G. 2 3
Rosanowski Sarah M. 4 5
Parkes Rebecca S.V. reparkes@cityu.edu.hk 1 2
1 Department of Veterinary Clinical Sciences, Jockey Club College of Veterinary Medicine and Life Sciences, City University of Hong Kong , Kowloon , Hong Kong , China
2 Centre for Animal Health and Welfare, Jockey Club College of Veterinary Medicine and Life Sciences, City University of Hong Kong , Kowloon , Hong Kong , China
3 Department of Infectious Diseases and Public Health, Jockey Club College of Veterinary Medicine and Life Sciences, City University of Hong Kong , Kowloon , Hong Kong , China
4 Digital Agriculture, Grasslands Research Centre, AgResearch Limited , Palmerston North , New Zealand
5 Equine Veterinary Consultants (EVC) Limited , Hong Kong , China
Vonk Jennifer
Electronic publication date: 2023 Aug 8
Publication date: 2023
Volume: 11
Electronic Location ID: e15747
Received 2023 Mar 20; Accepted 2023 Jun 22
Copyright: ©2023 Bukhari et al.
Copyright year: 2023
Copyright holder: Bukhari et al.
License: This is an open access article distributed under the terms of the Creative Commons Attribution License, which permits unrestricted use, distribution, reproduction and adaptation in any medium and for any purpose provided that it is properly attributed. For attribution, the original author(s), title, publication source (PeerJ) and either DOI or URL of the article must be cited.
License URL: https://creativecommons.org/licenses/by/4.0/

Keywords: Animal emotions, Animal pain, Animal sentience, Animal welfare, Donkey welfare, Owner behavior, Donkey working practices

Funding: City University of Hong Kong 9610463 This project was funded by the City University of Hong Kong (Grant Number 9610463). The funders had no role in study design, data collection and analysis, decision to publish, or preparation of the manuscript.

==============================
Working donkeys (Equus asinus) support human living standards globally. However, there is little information on the effect of human perceptions of emotion and pain on the welfare of working donkeys. We interviewed donkey owners (n = 332) in Pakistan to determine the relationship between human perspectives on donkey sentience: emotions and the ability to feel pain, and the routine working practices that could impact donkey welfare. The majority of donkey owners used padding under the saddle (n = 211; 63.6%; 95% CI (58.3%–68.9%)) and provided access to food (n = 213; 64.2%; 95% CI (58.9%–69.3%)) and water (n = 195; 58.7%; 95% CI (53.4%–64.1%)) during the working day. Owners reported that at some point in their donkey’s life, 65.3% (95% CI (60.2%–70.5%)) had load-associated injuries, of which 27.7% (n = 92; 95% CI (22.8%–32.5%)) were wounds, 20.5% (n = 68; 95% CI (16.1%–24.8%)) were lameness and 7.2% (n = 24; 95% CI 4.4%–10.0%) were back pain. In total, 81.3% (95% CI 77.1%–85.5%; n = 270) of owners believed that their donkeys felt pain, and 70.2% (95% CI (65.2%–75.1%; n = 233) of owners believed that their donkeys had emotions. Multiple correspondence analysis (MCA) was used to understand the relationship between owners’ recognition of emotions and pain in donkeys and their working practices. The MCA factor map revealed two clusters, named positive and negative clusters. The positive cluster included owner’s recognition of donkey pain and emotions, the availability of food and water, use of padding under the saddle, absence of injuries along with the willingness to follow loading guidelines. The negative cluster represented practices that did not benefit donkey welfare, such as using saddles without padding and a lack of food and water during work. The presence of injuries, owners not recognizing that donkeys feel pain and emotion along with an unwillingness to follow loading guidelines were also found in the negative cluster. We show that the owners who recognized sentience in their donkeys were more likely to use practices that are good for donkey welfare. The ability of owners to identify sentience in donkeys, along with their willingness to follow welfare guidelines, are important factors in improving the lives of working donkeys.

Introduction

Working donkeys have played a crucial role in the development of human civilizations and support some of the poorest communities in the world (Grace et al., 2022). There are approximately 50.5 million donkeys globally (Norris et al., 2021) and they support approximately 600 million people, including economically vulnerable communities in low and middle-income countries (LMICs) (Sommerville, Brown & Upjohn, 2018). Poor people in LMICs often depend on working donkeys for agriculture, construction, and the transportation of goods (Fig. 1) (Bukhari et al., 2022). However, until recently, little research has been conducted to assess and improve the welfare of working donkeys in LMICs. This could explain why their importance has frequently been neglected in government-level animal welfare regulations (Haddy et al., 2020). Hence, little is done to protect donkeys, resulting in compromised welfare as an outcome of harsh working conditions, exclusion in the legal system, and the disempowerment of both donkeys and donkey owning societies (Bukhari, McElligott & Parkes, 2021).

The most serious problems affecting donkey welfare include overloading and general overwork (Hameed, Tariq & Yasin, 2016; Bukhari, McElligott & Parkes, 2021). Increased load weights affect the health and welfare of donkeys (Bukhari, McElligott & Parkes, 2021). Moreover, unsuitable food and water, inappropriate saddling and harness, mishandling, hazardous practices, lack of supporting infrastructure (veterinary care, welfare legislation and regulatory bodies), and harsh environmental conditions are all factors that contribute to poor working donkey welfare (Birhan et al., 2014; Genetu et al., 2017; Rayner et al., 2018; Kamran et al., 2022), particularly for load-carrying donkeys.

Our perceptions of sentience (the ability to feel pain and to have positive and negative emotions (Proctor, Carder & Cornish, 2013)) shape our intentions, motivations, and behavior, which ultimately impact animal welfare (Luna et al., 2017; Luna & Tadich, 2019; Sinclair et al., 2022). The ability to feel pain and have positive and negative emotions is one possible definition of sentience that does not need the entity to be self-aware, but rather relies on their ability to experience internal psychological states (Proctor, Carder & Cornish, 2013). How owners treat their working animals is influenced by a complex combination of social convention, perceptions of sentience, economic constraints, and, in some cases, a lack of access to appropriate advice. Despite this complexity, one of the precursors to improving animal welfare is an acknowledgement of animal sentience (Luna et al., 2018). Human recognition of sentience improves overall animal welfare, health, and productivity (Budaev et al., 2020; De Waal & Andrews, 2022).

Figure 1 Two examples of loading practices in Pakistan (A) donkeys carrying bricks. (B) A donkey owner loading their animal with lucerne at a farm.

Photos: Syed Saad Ul Hassan Bukhari.

Understanding human-animal interactions (HAIs) is a vital component of any strategy aimed at improving the welfare of working animals (Spence, Osman & McElligott, 2017; Proops et al., 2018; Luna & Tadich, 2019). Animal welfare can be compromised by negative HAIs, which has adverse effects for the animal productivity, health, and wellbeing (Pinillos et al., 2016; Rault et al., 2020). Domestic animals, in addition to relying on people for food, frequently seek out positive social interactions with humans (Pinillos et al., 2016; Rault et al., 2020). Little is known about HAIs for donkeys and how they may affect their welfare. (Shah et al., 2019; Kamran et al., 2022). Increasing understanding of the human characteristics that influence owner-donkey interaction and how this affects donkey welfare could aid in the appropriate design of intervention strategies to improve the quality of owner-donkey relationships and the welfare of working donkeys (Luna & Tadich, 2019).

It is difficult to determine what donkey owners do (i.e., working donkey care and handling practices) and why they do it, unless welfare workers (non-government organizations (NGOs), government organizations, and veterinarians) take the time to speak with them. However, without this information any intervention will be based on incomplete and inaccurate knowledge of working animal welfare (Upjohn, Pfeiffer & Verheyen, 2014). There is little information on the relationship between human attitudes, empathy, recognition of emotions and pain, and donkey welfare. The objective of the current study was to investigate the relationship between human perspectives on donkey emotions and ability to feel pain along with willingness to follow loading guidelines, and routine working practices that could impact donkey welfare.

Materials & Methods

Study area and study design

The study areas and design have been described previously (Bukhari et al., 2022). Briefly, we conducted a cross-sectional survey of donkey owners in four different regions (Swat, Attock, Faisalabad, and Bahawalpur; Fig. 2) of Pakistan. Regions were selected due to different climatic and topographic conditions: mountainous, arid, irrigated plains, and sandy desert, respectively (Table 1) (Khan, 2021). These four regions cover 39,815 km2 of the country (approximately 4.5% of Pakistan).

Figure 2 Map of Pakistan showing the locations of the four study regions (sourced from ArcGIS).

Table 1 Elevation, coordinates and environmental conditions of the study areas in Pakistan (Khan, 2021).

Region	Elevation above sea level (m)	Coordinates	Highest average monthly temperature	Lowest average monthly temperature	Annual rainfall (mm)	
		Longitude	Latitude				
Swat	2591	72°54′	34°45′	37 °C in July	0 °C in January	1200–1400	
Attock	519	72°51′	32°55′	38 °C in June	3 °C in January	900–1000	
Faisalabad	185	73°08′	31°26′	41 °C in June	5 °C in January	300–400	
Bahawalpur	88	70°41′	28°39′	42 °C in June	4 °C January	100–150	

Questionnaire design

A questionnaire was designed to determine the relationship between human perspectives on donkey sentience, and routine working practices that could impact donkey welfare (Luna et al., 2017; Watson et al., 2020; Geiger et al., 2021). The questionnaire was developed based on local knowledge of the donkey’s routine work, with input from donkey owners and equine veterinarians (Bukhari et al., 2022). Due to low literacy rates, the questionnaire was conducted with owners verbally (Bukhari et al., 2022), and took around ten minutes to complete. All questions were close-ended, and the options were read to owners. In total, 332 donkey owners agreed to participate. In the first section of the questionnaire, the demographics of the owners and signalment (age, sex, breed) of the working donkeys were determined. Information on donkey loading practices was collected in the second part of the questionnaire. Regarding loading practices, we asked about donkey injuries, causes of load associated injuries, how owners assessed whether the load they are putting on their donkey is appropriate, whether padding is used under the saddle, the pace at which the donkey worked, the seasonal impact on workday duration, and the availability of feed or water during the working day. In the last section, with regards to the owner’s views on loading practices, we asked whether the current weight carried by their donkeys was appropriate, why people might overload their donkeys, whether they have noticed any changes to their donkey’s condition since purchasing it, whether they believed their donkey has emotions and whether they believed their donkey can feel pain. Finally, we asked owners whether they would follow loading guidelines for the benefit of their donkey, if such guidelines were available to them (Files S1 and S2).

Data collection

Interviewers verbally explained the study, its methods, and its purpose. The convenience sampling method was used, and owners were included based on their willingness to participate. The owner’s age was then determined verbally, and if they were over the age of 18, they were invited to be interviewed. Before the interview began, their informed verbal consent was obtained. After donkey owners agreed to participate, interviews were conducted using our questionnaire. The interviewers signed a “participant informed verbal consent form”. A third person signed the witness statement (witness, to ensure correct information exchange) on “participant informed verbal consent form” according to existing survey guidelines (Bukhari et al., 2022). Interviews were conducted in local languages (Urdu, Pashtu, Hindko, Pothwari, Punjabi, and Saraiki) after being translated by interviewers who were equine veterinarians fluent in both English and the respective regional local languages. This method was used to ensure maximum response accuracy while minimizing any confusion about the scientific terminology used in accordance with existing survey recommendations (Bukhari et al., 2022).

Variables

Variables collected during the interview are presented in Table 2. For ease of presentation, each variable has been assigned a short name and its structure (binary, categorical) is shown. For question, “Do you think your donkey has emotions?”, assigned short name is “Emotion”. For question, “Do you think your donkey feels pain?”, an assigned short name is “Pain”.

Table 2 Variables collected during the interview.

Variable collected	Assigned short name	Variable structure	
How you assess that the load you are putting on your donkey is practical for them?	Load assessment	Categorical	
Do you use padding under the saddle?	Saddle padding	Binary—yes or no	
How is the speed of loaded donkey selected?	Speed of donkey	Binary—chosen by donkey or triggered by you	
Does the duration of work per day vary by season?	Duration of work	Binary—yes or no	
Is there availability of feed during the working day?	Availability of food	Binary—yes or no	
Is there availability of water during the working day?	Availability of water	Binary—yes or no	
Have you seen load associated injuries in your donkey?	Injuries	Categorical	
What is the cause of load associated injuries?	Cause of injuries	Categorical	
The weight that you put on your animal, is it good for your donkey?	Appropriateness of current mounted weight	Categorical	
What is the reason that people overload their donkeys?	Reason for overloading	Binary—for more income or to finish work earlier	
Have you noticed an increase or decrease in general body condition since you bought this donkey?	Body condition	Categorical	
Do you think your donkey has emotions?	Emotion	Categorical	
Do you think your donkey feels pain?	Pain	Categorical	
Would you follow loading guidelines (if available) for benefit of your donkey?	Guidelines	Categorical	

Statistical analysis

Categorical data were described as counts, percentages, and with 95% confidence intervals (95% CI). A binary variable for the presence or absence of loads associated injuries (yes or no) was created. We investigated the relationship between owners’ recognition of emotion and pain in their donkeys, willingness to follow guidelines, and current working practices using multiple correspondence analysis (MCA) (Reid et al., 2017; Keogh et al., 2019). Variables spatially clustered together, were considered similar to each other, with spatial distance indicating rare associations (Greenacre, 2007; Alhuzali, Beh & Stojanovski, 2022). Where present, appropriate names were generated to describe clusters. The MCA included region, area, injuries, saddle padding, speed of donkey, availability of feed or water. Guidelines, emotion, and pain were also included in MCA. Variable counts, percentages and 95% CI was computed using statistical package for social science (SPSS) version 25.0, whereas, MCA was conducted using the open source software RStudio-2022.07.1-554 (RStudio, 2022).

Ethical approval

The research was approved by the Human Subjects Ethics Sub-Committee, City University of Hong Kong (Approval reference no. JCC2021AY003).

Results

Loading practices

Owners reported that they assessed the practicality of the weight of the load for donkeys by weighing the load (n = 48; 14.5% (95% confidence interval (CI) 10.7%–18.2%)), or by observing their donkey’s behavior (n = 84; 25.3% (95% CI 20.6%–30.1%)). The majority of the donkey owners used padding under the saddle (n = 211; 63.6%; 95% CI (58.3%–68.9%)); and provided access to food (n = 213; 64.2%; 95% CI (58.9%–69.3%)) or water (n = 195; 58.7%; 95% CI (53.4%–64.1%)) during the working day (Table 3).

Table 3 Practices of working donkey owners (n = 332) related to mounted load carrying for working donkeys in Pakistan.

Variables	Categories	Number	Percentage (%)	95% confidence interval (CI) lower—95% CI upper (%)	
Load assessment	By weighing load	48	14.5	10.7–18.2	
Checking donkey behavior	84	25.3	20.6–30.1	
Adding load approximately	178	53.6	48.2–58.9	
I don’t know	22	6.6	3.9–9.3	
Saddle padding	Yes	211	63.6	58.3–68.9	
No	121	36.4	31.2–42.7	
Speed of donkey	Chosen by donkey	90	27.1	22.3–31.9	
Triggered by the owner	242	72.9	68.1–77.7	
Duration of work	Yes	201	60.5	55.2–65.8	
No	131	39.5	34.1–44.7	
Availability of food	Yes	213	64.2	58.9–69.3	
No	119	35.8	30.6–41.0	
Availability of water	Yes	195	58.7	53.4–64.1	
No	137	41.3	35.9–46.6	

Injuries associated with mounted loads

Owners reported that 65.3% (95% CI 60.2%–70.5%) of their donkeys had load-associated injuries at some point of their life, of which 27.7% (n = 92; 95% CI 22.8%–32.5%) were wounds and 20.5% (n = 68; 95% CI 16.1%–24.8%) were lameness. The majority of donkey owners believed that overloading was the cause of load associated injuries (n = 228; 68.7% (95% CI 63.6%–73.7%)), while others believed that type of load (n = 34; 10.2% (95% CI 6.9%–13.5%)) or practices during loading and unloading (n = 70; 21.1% (95% CI 16.7%–25.5%)) were the cause of injuries (Table 4).

Table 4 Load associated injuries and their possible causes reported by owners (n = 332) for working donkeys in Pakistan.

Variables	Categories	Number	Percentage	95% confidence interval (CI) lower–95% CI upper (%)	
Injuries	Wounds	92	27.7	22.8–32.5	
Lameness	68	20.5	16.1–24.8	
Back Pain	24	7.2	4.4–10.0	
Wounds and lameness	21	6.3	3.6–8.9	
Wounds and back Pain	5	1.5	0.0–3.6	
Wounds, lameness, and back Pain	7	2.1	0.2–3.7	
No injuries observed	115	34.7	29.5–39.8	
Cause of injuries	Type of load	34	10.2	6.9–13.5	
Overload	228	68.7	63.6–73.7	
Practices of loading and unloading	70	21.1	16.7–25.5	

Owners’ views on loading practices

Almost half of the owners (n = 151; 45.5% (95% CI 40.1%–50.8%)) believed that the weight their animals carried was appropriate. In total, 81.3% (95% CI 77.1%–85.5%; n = 270) of owners reported that they believed their donkeys felt pain and 70.2% of (95% CI 65.2%–75.1%; n = 233) owners believed that their donkeys had emotions. A total of 190 (57.2% (95% CI 51.9%–62.6%)) donkey owners said they would be willing to follow loading guidelines (if available) for the benefit of their donkey (Table 5).

Table 5 Owners’ (n = 332) views on loading practices related to mounted load carrying by working donkeys in Pakistan.

Variables	Categories	Number	Percentage	95% confidence interval (CI) lower–95% CI upper (%)	
Appropriateness of current mounted weight	Yes	151	45.5	40.1–50.8	
No	69	20.8	16.4–25.2	
I don’t know	112	33.7	28.6–38.8	
Reason for overloading	For more income	183	55.1	49.7–60.5	
To finish work earlier	147	44.3	38.9–49.6	
Both for more income and to finish work earlier	2	0.6	0.0–1.4	
Body condition	No change in body condition	221	66.6	61.5–71.7	
Body condition increased	49	14.8	10.9–18.6	
Body condition decreased	62	18.6	14.4–22.9	
Emotion	Yes	233	70.2	65.2–75.1	
No	62	18.7	14.4–22.9	
I don’t know	37	11.1	7.7–14.5	
Pain	Yes	270	81.3	77.1–85.5	
No	45	13.6	9.8–17.2	
I don’t know	17	5.1	2.7–7.5	
Guidelines	Yes	190	57.2	51.9–62.6	
No	8	2.4	0.7–4.1	
I don’t know	134	40.4	35.1–45.7	

Multiple correspondence analysis of working practices and recognition of emotion and pain in donkeys

The MCA factor map revealed two clusters of variable categories, named the positive and negative cluster. Recognizing donkey pain or emotion, a willingness to follow loading guidelines, availability of feed and water, absence of injuries, and use of saddle padding were grouped in the positive cluster. In contrast, the negative cluster represented practices that did not benefit donkey welfare, such as the use of saddles without padding and a lack of feed or water during work. The presence of injuries, owners not recognizing that donkeys feel pain or emotion along with unwillingness to follow loading guidelines were also found in the negative cluster. Speed of the working donkeys did not fit into the clusters (Fig. 3).

Figure 3 Multiple correspondence analysis (MCA) factor map with respect to categorical variables.

Variance explained by dimension one and dimension two was 45.89% and 22.09%, respectively. Color dots represent categorical variables, and signs (+, −, ±, S, A, F, B, R, PU, U) represent their categories.

Discussion

We studied the relationship between human perspectives on donkey emotions and the ability to feel pain, and the routine working practices that could impact donkey welfare. This is the first report to elucidate owners’ perceptions of donkey sentience and how it affects the lives of working animals in Pakistan. Owners who recognized sentience in their donkeys were more likely to apply good welfare practices, such as willingness to follow loading guidelines, using padding under the saddle, and providing food and/or water during the working day (positive cluster, Fig. 3). These owners also stated that their donkeys were uninjured during work. Owners who did not recognize sentience in their donkeys were more likely to employ practices that were detrimental to donkey welfare, such as using saddles without padding and a lack of food and water during work (negative cluster, Fig. 3). Owners reported that at some point in their donkey’s life, most had load-associated injuries. Human empathy, emotions toward animals, and perception of animal pain contribute to better HAIs and can enhance the welfare of working equids (Lanas, Luna & Tadich, 2018; Proops et al., 2018; De Waal & Andrews, 2022). Thus, we show that improved working donkey welfare occurs when the perception of sentience is widely held by owners.

We showed owners’ perceptions of emotions and pain are clustered with various positive welfare practices (Positive cluster, Fig. 3). Previously, when owners shared an affective perception, a favorable welfare status (low occurrence of depressed working animals) was observed (Luna et al., 2017). It has also been reported that when owners believe their animal has feelings and needs, working animals have better welfare (for example, owners offer sufficient food and water to donkeys) (Luna & Tadich, 2019). Empathy for animals fosters positive attitudes toward animals and heightens ability to acknowledge animal pain (Lanas, Luna & Tadich, 2018; Proops et al., 2018; De Waal & Andrews, 2022), which may lead to better care of the donkeys by their owners, ultimately benefiting welfare of working donkeys. Every interaction and experience an animal receives throughout its life leads to a negative or positive response, impacting the welfare of that animal (Wolfensohn, 2020). Little research has been conducted to identify the human characteristics that modulate the owner-donkey interaction and how these may affect the welfare of working animals (Shah et al., 2019; Kamran et al., 2022).

In our survey, nearly two-thirds of owners reported their donkey experiencing load associated injuries (wounds, lameness, and back pain). This is similar to a study conducted in India on donkeys, where 62.8% of the population had injuries, with wounds comprising most injuries (Rayner et al., 2018). Similarly, wounds were highly prevalent (72.1%) among the working equines in Ethiopia (Biffa & Woldemeskel, 2006). In comparison to this, the prevalence of wounds in the Merawi-Ethiopia (38.3%) was quite low (Tsega et al., 2016), and in Baluchistan-Pakistan the prevalence of wounds and lameness was 9.2% and 16.3%, respectively (Kamran et al., 2022). However, it is not clear whether these donkeys were doing load associated work in those regions. Moreover, a recent study of working donkeys pulling carts in Faisalabad-Pakistan discovered that 96% of donkeys were lame (which is a very high number) when examined by a veterinarian, despite the fact that the donkey was still in harness (i.e., donkeys were examined while the harnessing system was attached) (Khan et al., 2022). The lack of owner reported injuries was clustered with recognition of emotion and pain (Positive cluster, Fig. 3) in our report. In Baluchistan-Pakistan, 86.2% of donkey owners do not whip their donkeys to avoid injury and report that animals feel pain in the same way that humans do (Kamran et al., 2022). This pertains to the fact that when owners believe their donkey feels pain, their animals have a low prevalence of injuries.

Most donkey owners in this study used padding on their load carrying donkeys. The use of saddles with padding was clustered with the recognition of emotion and pain (Positive cluster, Fig. 3). Previous studies have identified that appropriate saddle padding was an important positive welfare practice for working donkeys (Birhan et al., 2014). In Ethiopia, the prevalence of back sores was associated with saddle condition, and donkeys used with inappropriate saddle were twice as likely to have back sore than those used with the proper saddle (Birhan et al., 2014). Wounds caused by inappropriate saddle were higher in both working horses (62.7%) and donkeys (50.6%) in Ethiopia. This could be due to the pressure, friction, and shear lesions caused by saddles without adequate padding. Furthermore, in Ethiopia, 76% of horses and 89.7% of donkeys were used for continued work despite the presence of wounds (Genetu et al., 2017). Working animals with inappropriate saddles had higher prevalence (63.3%) of wounds in Southern Ethiopia (Tesfaye, Deressa & Teshome, 2016), Eastern Ethiopia (40.9%), Morocco (54%) and up to 45% wounds are related to the saddle in the Egyptian brick kilns (Farhat, McLean & Mahmoud, 2020). A possible explanation for these observations is a lack of understanding of sentience, basic husbandry practices, and donkey needs.

Overall, 64.2% and 58.7% of owners reported feeding and watering their donkeys during the working day, respectively. In Ethiopia, 72.5% donkey owners practice feeding twice per day, and they provide feed before loading (Tesfaye, Deressa & Teshome, 2016) and in Chile, most working animals (83%) were fed three times per day (Luna et al., 2017). In Ethiopia, food shortage is the major constraint to productivity and work performance of donkeys (Birhan et al., 2014; Tsega et al., 2016). They are forced to work without proper feed, reflecting their poor welfare status (Birhan et al., 2014; Tsega et al., 2016). Moreover, in Ethiopia, 85% donkey owners provide water three times per day, regardless whether or not donkey was working (Tesfaye, Deressa & Teshome, 2016). However, there are reports of animals carrying a pack or pulling a cart for 8 h without water (Pritchard et al., 2005). Long working hours without proper watering not only cause depression in donkeys, but can also result in death in severe cases (Hameed, Tariq & Yasin, 2016). In contrast, in Chile, the majority of working animals (90%) had access to drinking water throughout the day as owners believe that their animal has feelings and needs (Luna et al., 2017).

Body condition is an important measure of donkey welfare (Haddy et al., 2021). Most of the owners (66.6%) reported no change in perceived body condition, and 18.6% of owners reported that the body condition of their donkey had decreased since they bought it. In previous reports, 41.6% of Ethiopian (Moltumo et al., 2020), and 56% Egyptian working donkeys had a poor body condition (Farhat, McLean & Mahmoud, 2020). However, it is essential to note that in Ethiopia and Egypt, body condition was measured by trained professionals, in contrast to our study, in which perceived changes were reported by owners rather than direct evaluations of body condition. Donkeys have been documented to have a poorer BCS than mules and horses in LMICs (Burn, Dennison & Whay, 2010). This could be due to a number of factors such as poor management, high work load, shortage of essential nutrients, scarcity of feed, and lack of supplementary diets (Herago et al., 2015; Bukhari, McElligott & Parkes, 2021). Poor body condition and overwork are the main contributors to the occurrence of wounds in working donkeys (Tesfaye, Deressa & Teshome, 2016). In Mexico, donkeys with poor body condition were more likely to acquire wounds due to load associated work (Tesfaye, Deressa & Teshome, 2016; Haddy et al., 2021). This may be because these low BCS donkeys have less natural padding to protect themselves from injuries associated with mounted load carrying. Almost half of our study population was associated with rural agricultural business. These donkeys are not subject to a heavy workload (Bukhari et al., 2022) in comparison with donkeys working at brick kilns, so the reported positive changes in BCS may be linked to workload.

Owner acquisition of empathic abilities towards their animals and the establishment of positive HAIs can have important consequences both for the performance and welfare of working equids (Luna & Tadich, 2019). For example, owners with a greater perception of sentience in equines kept animals in a better welfare state and may explain why there was high frequency of horses in Chile that responded positively to both the observer and the owner (Luna et al., 2017; Kamran et al., 2022). In addition, owners’ perceptions of pain towards equines were found to be highly associated with their empathic skills (Luna et al., 2018). Therefore, strategies intended to improve the welfare of working equids should not only consider the identification of their main welfare problems, but should also include the assessment of the main factors that modulate HAIs from the human perspective, such as the owner’s empathy towards animals (Tadich et al., 2016; Proops et al., 2018; De Waal & Andrews, 2022). The motivational bases that underpin attitudes toward animals must be identified in order to develop strategies to improve donkey welfare. It is important to understand that working equids face different challenges in different communities and geographic sites (Haddy et al., 2020).

Our research represents a snapshot of some working practices and their association with owner’s thinking of donkey sentience in three regions of Pakistan. Future surveys should investigate welfare parameters such as proper housing, access to veterinary care after work-related injuries, and other husbandry practices in relation to owner perception of sentience. Studying perception of working donkey emotions and pain is important (Lanas, Luna & Tadich, 2018) because knowledge about how working donkey owners perceive emotions and pain in their animal, and how recognition of sentience may influence the welfare of these animals, is scarce. Interviewer translated owners’ interviews and the questionnaire in the local language verbally during the donkey owner’s interviews, this may lead to the loss of meaning or alteration in different languages with different interviewers/translators (Filep, 2009). Moreover, we do not yet know how much a donkey should safely carry and what the loading limits/guidelines for donkeys should be. This may affect the owners’ answer for “would you follow loading guidelines?” However, their answers to the question do show their intent to follow scientific guidelines for improved welfare for their animal. In addition, we should also investigate donkey behavior in working setup and their association with owner’s attitude towards donkey sentience and their welfare. More targeted interventions to improve welfare will be possible with a better understanding of the working donkey-owner relationship.

Conclusions

Empathy, attitude towards animals, and pain perception are some of the human psychological traits that influence human-animal relationships and animal welfare. The ability of owners to identify sentience in donkeys is an important factor for improving welfare and may influence how working animals are treated. We show that the owners who recognized sentience were more likely to use practices that are good for donkey welfare (positive cluster) in a challenging working environment. Explaining donkey sentience to owners (not following good working practices (negative cluster)) and the benefits of improving their donkeys’ welfare through proper working practices will help motivate positive change.

Supplemental Information

File S1 Raw data

Click here for additional data file.

File S2 Study questionnaire

Click here for additional data file.

Additional Information and Declarations

Competing Interests

Author Contributions

Human Ethics

Field Study Permissions

Data Availability

Alan G. McElligott and Rebecca S. V. Parkes are Academic Editors for PeerJ. Sarah M. Rosanowski is an independent epidemiological consultant who consults under Equine Veterinary Consultants (EVC) Limited and Digital Agriculture, Grasslands Research Centre, AgResearch Limited. The remaining authors declare that the research was conducted in the absence of any commercial or financial relationships that could be construed as a potential conflict of interest.

Syed S.U.H. Bukhari conceived and designed the experiments, performed the experiments, analyzed the data, prepared figures and/or tables, authored or reviewed drafts of the article, and approved the final draft.

Alan G. McElligott conceived and designed the experiments, authored or reviewed drafts of the article, and approved the final draft.

Sarah M. Rosanowski analyzed the data, authored or reviewed drafts of the article, and approved the final draft.

Rebecca S.V. Parkes conceived and designed the experiments, authored or reviewed drafts of the article, and approved the final draft.

The following information was supplied relating to ethical approvals (i.e., approving body and any reference numbers):

The Human Subjects Ethics Sub-Committee, City University of Hong Kong (Approval reference no. JCC2021AY003) approved the study.

The following information was supplied relating to field study approvals (i.e., approving body and any reference numbers):

Data collection was conducted with the permission and cooperation of the regional veterinary service.

Moreover, Syed S.U.H. Bukhari is from Pakistan, and doing PhD is Hong Kong, and Corresponding author of the study.

The following information was supplied regarding data availability:

The raw data are available in the Supplemental Files.

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
