# Peer review of "Recognition of emotion and pain by owners benefits the welfare of donkeys in a challenging working environment"

_PeerJ, doi:10.7717/peerj.15747_

## Round 0.1 · original submission · Minor Revisions

This work clearly addresses an extremely important topic. The rationale and basic design are sound. Both reviewers also see the merit in the work but each have some suggestions to improve clarity.

I have a few additional comments of my own.

First, the sample size is impressive, especially considering the interview method.
You clearly define sentience on line 72. However, for some, the definition is stricter and requires a sense of self-awareness. It may be important here for clarity to emphasize that this is one possible definition of sentience that does not require the being to be self-aware but instead focuses on their ability to experience internal psychological states.

The “on the fly” translation method is not ideal and may lead to loss of meaning or alterations in different languages and with different interviewers and translators so this should be addressed in the limitations.

Because I suspect many other readers will not be fluent in MCA, can you please provide a rationale for this approach instead of using a typical regression or SEM analysis?

Be careful to note that you measured only the owners’ perceptions of their donkeys’ well-being. You did not directly measure the condition or welfare of the donkeys. You do note this around lines 275-277 but be sure to describe as “owner perceived donkey condition” rather than “donkey condition” throughout or note it once at the outset for transparency.

Small edits
Line 116, missing “the” before “first section.”
Line 117, what is “signalment?”
Line 168 missing “of” before “the donkey.”
Line 220, acknowledge does not need to be capitalized.

Reviewer 1 ·

Basic reporting

This is a novel manuscript describing the relationship between the ability of donkey owners to recognise sentience and some welfare outcomes in therio donkeys. The authors have included the relevant literature available. I believe some parts of the discussion may be improved, I will detail them, but in general it is a clear manuscript.

Experimental design

The manuscript is within the scope of the journal. The research question/aim is clearly defined at the end of the introduction (lines 96-99) and is a relevant one since it is a first approach to understanding how perceptions on sentience by donkey owners can affect the welfare of their animals. Although this has been studied for other equids it is a first report for donkeys.
In relation to the methods it would be nice to have more information about the questionnaire in the text (lines 110-116). For example, I think it should be stated in this part that all questions were close-ended and that the options were read to owners, I understand according to the attached questionnaire that this is how it was applied, or did owners provide an open answer and the researcher selected an option according to the answer?
For the question "would you follow loading guidelines....? how do you think that the low literacy rates would affect the answer? maybe something about this should be further discussed in the discussion section.
For the analysis an MCA was used, I am not very familiar with this type of analysis but it seems that there might be an overinterpratation of the results. The authors describe two clusters (a positive and a negative one) but for both clusters there are variables on both sides of the 0 axis and also some variables seem to be very close to the origin, being less discriminating, without the table with the raw data used for tha analysis it is difficult to do a correct interpretation. Maybe it should be considered to perform a multinomial logistic regression to understand which management practices, or which welfare outcomes are explained by the owners perception of sentience.

Validity of the findings

The manuscript is novel and all underlying data has been provided. As mentioned earlier I have some doubts with the MCA interpretation, but again I would like to be clear that it is not a methodology I am very familiar with.
In general, through the discussion each finding is dicussed separately and not in terms of the two clusters described, it would be worth providing a discussion about the clusters, what do owners from each cluster share? why is it relevant to have these clusters?
In the conclusions section I would try to incorporate the main finding of the study associated to the clusters that have been described.

Additional comments

Minor additional comments:

Line 204: Please check the sentence "treating donkeys better leading to lack of injuries"
L220: acknowledge
L236: "the fact that the donkey was still in harness", do you mean that donkeys were examined while the harnesing system was attached? would you expect that 96% to be higher or lower?
L239-241: Did the other studies assess sentience recognition? forme it seems that it cannot be concluded that owners that recognize donkey sentience use positive welfare practices by only providing prevalences of wounds and lameness.
L246 instead of insufficient saddle y recommend using: donkeys used inadequate/inappropiate saddle types/design or poorly fitted saddles, if this is what is meant, or do the authors mean insufficient padding material?
L250 please check the grammar of 89.7% of donkeys used for work continuously...
L262-263: please check this sentence: 2016). They are made to work without adequate feed indicating their poor welfare status.
References:
Lines 369 and 372 the reference Hameed et al 2016 is duplicated but with letters a and b, please check this
Line 380: Equus asinus not ascinus
Line 390 and 393: The reference Lanas et al 2018 is duplicates with letters a and b please check this
Line 438 the first name of co authors are included, only the initial should be included

Tables: I suggest including the number of owners in the title of tables

·

Basic reporting

L57: Only working donkeys? I would argue for working equids (horses, mules, hinnies and donkeys). Especially since you mention equids in line 67. Perhaps start of with the general statements regarding equids, followed by narrowing it down to donkeys.
L69: It is not clear what “lack of supporting infrastructure” means to donkey welfare? Would this rather be related to the donkey owners?
L86-87: The sentence “Research to understand HAIs for donkeys and how these may affect their welfare is new” is not clear to me – do you mean that this field of science/research is new? If you mean that little is known, please rephrase, and add references to back up the statement if it implies that research is present (not entirely lacking).
L92-93: Donkey owners do what? The context is missing.
L116: Were these 332 owners blind to the aim of the study? I mean, could their prior knowledge about the study have affected their responses?
L117: what is meant with the donkey’s "signalment"?
L130: Okay this answers my comment for L116, but could this have affected your results? I mean, could some respondents have answered more positively in order to “do right” for their animals/for the purpose of the study, unintentionally?

Experimental design

Section data collection: Very nice to see such good data collection ethics!
L144: This short section comes across a bit too short. I suggest to briefly mention your variables (or themes of variables), or at least the most important ones, to allow a good text flow.
L158-159: Just a reminder of citing any R-packages you may have used too.
L160: Perhaps a few words describing the ethical considerations feeding into your ethical application? I’m aware that supplementary files are available, but more information here could help ensure all readers get an insight to these important considerations.

Validity of the findings

Please see my comments under 1. Basic reporting.

Additional comments

L216-219: Very long sentence, I suggest shortening to ease readability.
L220: Change “Acknowledge” to acknowledge.
L223-225: Again, as in the introduction, cite the studies that are done, if any.
L242: Add “in this study” after “owners
L243-245: Add references to this claim.
L360: Please do not use citations in a conclusion – a conclusion is concluding remarks on your results, not the results of others.

Thank you for an interesting read!

---

## Round 0.2 · Minor Revisions

Thank you for the efforts you have undertaken to improve the clarity of your manuscript. I have a few additional requests before I can accept the paper.
It would help to have clear hypotheses at the end of the introduction such as that owners who believe that donkeys feel pain may be more likely to follow guidelines etc. Make it clear how each analysis is testing a particular hypothesis. That will help make it clear what you are aiming to accomplish with the MCA. Could you include a bit of your response to my question about why you used an MCA in the actual paper please?

Under “Loading Practices,” you have only 132 participants reporting on how they assessed the practicality of the weight (or 39.8%). Where are the data on the missing 60.2% of respondents?

The subheadings don’t seem appropriate. For example, access to food and water doesn’t seem to belong under this subheading of Loading Practices. The owners’ views on loading practices could be under the main Loading Practices subheading. Why is the information about whether the donkeys feel pain and have emotions under this section?

Did each owner report on only a single donkey or could they be reporting on multiple donkeys? That is, is it accurate to state (line 180) that 65.3% of donkeys had injuries or that 65.3% of owners that you interviewed reported that at least some of their donkeys had injuries of various types? Be sure to be very precise in how you describe the results.

Line numbers refer to the reviewing PDF.
Line 58, place a comma after “until recently.”
Line 96 and check throughout; place , after i.e. and e.g.
Line 118 there is an extra space after the citation before the .
Line 226, and 272 delete the “.” Before the citation.
Line 228, it would be good to add another clause “which may lead to better care of the donkeys by their owners, which benefits welfare….
Line 246, refer to “owner reported injuries” rather than “injuries.”
Line 256 “back sores” not “sore.”
Line 291, wound should be wounds
Line 299, 305, do you mean “empathetic” rather than “empathic?”
Line 320 reword “Interviewers translate.. owners’ interviews. This may lead…
Line 322, delete the “as”
Line 324 “owners’” needs an apostrophe.

Reviewer 1 ·

Basic reporting

The manuscript is clear and the references are sufficient to provide field background

Experimental design

The authors have included an important number of donkey owners to their investigation, which is the first one to investigate donkey owners perception of pain and emotions of their animals.

Validity of the findings

The results are novel and meaningful to the field

Additional comments

The authors have addressed all my comments and I am happy with the changes done to the manuscript. I have some minor suggestions:
Line 98: by welfare workers, do you mean people working for NGOs? please clarify
Line 391. please use the complete journal name
Line 490. Instead of Aline A, use de Aluja, A.

---

## Round 0.3 · accepted · Accept

Thank you for these final clarifications and corrections.